# Association of company downsizing with the risk of mental distress among employees in the Norwegian working population

**Eirik Degerud** *, **Andrea R. Marti, Tom Sterud**

National Institute of Occupational Health, Oslo, Norway

* Eirik.degerud@stami.no

## Abstract

Studies indicate that downsizing can have a negative influence on the mental health of employees who keep their jobs. This study examines how downsizing impacts the mental health of remaining employees, focusing on their proximity to downsizing and ratings of involvement, information, and support during downsizing. Based on data from three iterations (2013, 2016 and 2019) of a nationwide survey in Norway, we included respondents employed by the same company in two consecutive surveys. We used logistic regression models adjusted for age and sex to calculate odds ratios (ORs) with 95% confidence intervals (95% CIs) for incident mental distress in the second survey among employees without distress in the first survey. Compared to unexposed employees ($n = 2,571$), the OR (95% CI) for employees exposed to downsizing was 1.57 (1.16 – 2.13) overall ($n = 1,300$). When differentiating exposed employees by proximity, the ORs were 1.78 (1.26 – 2.49) for those with downsizing in their own department ($n = 784$) and 1.26 (0.79 – 1.95) for those with downsizing in another department ($n = 516$), respectively. Furthermore, among employees exposed in their own department, the ORs differed according to their rating of involvement, information, and support during downsizing: 1.27 (0.82 – 1.91) for sufficient ($n = 580$) and 5.02 (3.03 – 8.04) for insufficient ($n = 149$). Downsizing was associated with a higher risk of mental distress among employees who kept their job, but the findings also suggest that companies might reduce this risk through employment involvement, information and support to employees affected the most by downsizing.

## Introduction

Organizational restructuring is a broad concept encompassing various forms of change within a company and downsizing [1] is one of many activities or events that can lead to restructuring. Downsizing is defined as a strategic management tool used by organizations to become more efficient, competitive, or profitable [2] by means of reducing personnel. Personnel reduction can be achieved by reducing work hours,

**Data availability statement:** The data used in this study was obtained from a third party and we cannot legally distribute or share the data. Please read this statement on how to access the data in full via the third party. A researcher that obtains the data should be able to replicate the findings in the study by following the information outlined in the Method section. The data used in this study was obtained from The Norwegian Agency for Shared Services in Education and Research. Their Survey Bank contains research data from the social sciences, humanities, and medical and health research (https://sikt.no/en/tjenester/finn-data/survey-bank), in total 750 000 questions from 3000 surveys going back to 1957. Some of the data available through the Survey Bank can be accessed by everyone and is available for downloading directly. The remaining data in the Survey bank can be accessed by ordering the data, but only by applicants that meet the specific requirements for accessing data from that survey. The guide to ordering data from the Survey Bank (https://sikt.no/en/tjenester/finn-data/guide-ordering-data-sikt) includes the following steps: the applicant needs to create an account, find the relevant data and add it to the shopping cart, create a project, fill in institution responsible for the project, enter the purpose of the data order, and sign the user agreement. In this study, we ordered a file containing panel data from three iterations (2013, 2016 and 2019) of The Norwegian Survey on Living conditions—working conditions surveys (https://surveybanken.sikt.no/no/study/24946cf2-9b34-4ea7-9c08-82b48c2c6286/undefined?type=studyMetadata&file=f4ba3791-17f2-4278-96dc-9b522a-73b3a7/3). The Survey Bank has the following data availability statement for this data: «Data are made available for research, students and teaching at approved institutions by The Research Council of Norway or Eurostat when ordered from the Survey Bank. In addition, the following bodies may be granted access to data for the production of statistical results and analyses a) Central government bodies, county authorities and municipal authorities b) The Norwegian bank c) The Office of the Auditor General of Norway d) National statistical offices in other countries e) International organizations on the basis of an agreement under international law». This means that the data can only

natural attrition, layoffs, or a combination [3]. Downsizing causes unemployment, which can have severe consequences for those afflicted [4–7].

A growing body of research has examined the implications of restructuring and downsizing for the well-being and mental health of employees that keep their job, known as 'survivors'. A systematic review of studies published between 2000 and 2012 by De Jong *et al.* found that most studies reported a sustained negative association between restructuring activities and well-being [8]. Bamberger *et al.* reviewed studies on the impact of organizational change on mental health. Their findings suggested that the adverse effects of restructuring on mental health might be short-term, primarily arising from the initial shock of the event, but their findings were mostly based on cross-sectional research [9]. The studies comprised a Finnish cohort of public sector employees amidst the 1990s economic recession, which unveiled increased instances of sick leave [10–12], disability pension [13] and psychotropic drug use [14] among downsizing survivors. Comparable results were later on reported from a US cohort of continuously employed workers in a multi-site manufacturing firm during the Great Recession of 2009. Workers in high-layoff plants reported higher levels of work stress [15] and had a higher risk of developing hypertension [16] than workers in low-layoff plants. Additionally, prolonged follow-up of these workers revealed a more pronounced increase in psychotropic drug use and mental health related prescriptions among workers in high-layoff plants [17], but no difference was found in depression risk [16] or healthcare service utilization [18]. Two studies of the Norwegian general working population provided mixed results, with a questionnaire-based measurement of downsizing [19] being linked to increased mental distress risk in one study, whereas a registry-based measurement [20], was not linked to a higher risk in the other study. Overall, longitudinal studies suggest a generally negative impact of downsizing on mental health, although further research is needed to clarify specific mechanisms and moderating factors.

Several mechanisms could explain an association with mental health. The process of losing colleagues can be painful by itself. Furthermore, downsizing may be perceived as a threat and trigger feelings of uncertainty about the future job situation [21,22], which in turn can affect mental health [23]. Job insecurity is one of the main proposed mediators of how downsizing and other restructuring activities influence mental health [9,24–26]. It may also introduce what may be defined as positive changes in working conditions including new tasks, opportunities and responsibilities [24,27], but on the other hand increase work load, instability, and job strain [2].

Research has highlighted that the way restructuring is conducted plays a crucial role in shaping its impact on employee well-being and that negative effects can be mitigated when employees are actively involved in the change process or have a role in decision-making. This includes whether the company provides information and involve employees in the transformation of the workplace, which could provide a sense of control over the situation [28], alleviate feelings of uncertainty [22] and mitigate a potential negative effect on mental health [29]. The review by De Jong *et al.* identified seven studies emphasizing the importance of employees' perceptions of the restructuring process [8]. Factors such as transparent communication, procedural justice, and change-related training were reported to positively influence well-being.

be made available by ordering the data from the Survey Bank and provided that the applicant meets the criteria. The National Institute of Occupational Health in Norway is listed as a research institution by the Research Council of Norway and we could therefore order data. For international researchers, the institution needs to be recognized as a research entity by Eurostat. An extensive list of international organizations currently recognized by Eurostat as a research entity can be found here (https://ec.europa.eu/eurostat/web/microdata/overview). The list includes universities, research institutes or research departments in a public administration, banks and statistical institutes. It also contains information on how to apply to have a research organization recognized as a research entity, a procedure that according to Eurostat takes around 4 weeks. When this is in order, the researcher should be able to order data from the Survey Bank at The Norwegian Agency for Shared Services in Education and Research on the same level as we the authors.

**Funding:** The authors received no specific funding for this work.

**Competing interests:** The authors have declared that no competing interests exist.

Related findings were also reported in cross-sectional data from European surveys, where information and employee involvement or consultation were identified as key factors that mitigated the negative effects of downsizing on health and well-being [24]. Other work-related factors, such as perceived fair treatment, leadership, and social support, were also found to be potential mitigating factors. However, these factors were generally assessed and not directly linked to the respondents' experience with how their organization managed downsizing [24]. These findings suggest that planned organizational efforts, such as effective communication strategies and fair restructuring procedures, can help mitigate the negative impact of downsizing. However, there is a lack of prospective studies that evaluate the impact of downsizing on mental health in general working populations, particularly regarding whether planned organizational efforts might effectively reduce these negative effects.

This study contributes to the research on the specific association between downsizing and mental health by employing longitudinal data, thus reducing the risk of reverse causation. Using a validated measure of mental distress and data from a representative sample of the general working population, we examine not only the overall impact of downsizing but also variations in exposure within companies. By differentiating employees based on their proximity to the downsizing event within the company and their perceptions of involvement, information, and support, this study aims to shed light on the role of organizational responses in mitigating the potential negative effects of downsizing on employee mental health.

## Methods

### Source population, study design, and participant selection

The Survey of Living-Working Conditions is an ongoing triannual survey. Each iteration invites a representative sample of Norwegian residents aged 16–66 years to attend. A sample from 2006 (n = 18,679) was re-invited in full in 2009 and 2013, but in 2016 and 2019, one third (n = 6,333) was rotated out and replaced. It obtains data from a questionnaire filled out by a trained interviewer during a phone interview and from registries. Statistics Norway conducts the survey according to statutory rules and participants give their informed consent to the use of their data in official statistics and research based on anonymised data. The study did not require approval by an Ethics Committee because the data areanonymous.

This study is based on a panel file of respondents who attended in 2013, 2016 and 2019 [30]. The first survey (April – January 2013/14) invited 20,492 individuals and 10,875 (53.1%) attended. The second (September – April 2016/17) invited 20,272 individuals and 10,655 (52.6%) attended. The third (August – March 2019/2020) invited 19,678 individuals and 11,212 (57,1%) attended. The total number of responders was 32,752 nested in 22,573 individuals.

To mimic a target population defined as employees in the general working population, we defined the source population as survey respondents in paid work for at least one hour or temporarily absent from work during the interview week and who were employed in a company in the private or public sector and not self-employed. In this source population, we identified respondents who participated in two consecutive

surveys, hereafter referred to as the baseline and the follow-up survey (conducted in 2013 and 2016; 2016 and 2019, respectively). To be eligible, respondents also had to retain their status as an employee in the same company at the two consecutive surveys, and the company a size of at least ten people at baseline. Furthermore, respondents with mental distress at baseline were excluded. Lastly, each respondent had to have data on downsizing, mental distress, and important covariates. Individuals attending all three surveys could be included twice if all inclusion criteria were met.

## Downsizing

A dichotomous ("No", "Yes") downsizing exposure variable and a three-level categorical exposure variable, that included information on proximity to downsizing, was obtained with the question "During the past three years, has the company where you now work performed downsizing?". Response alternatives were "No downsizing", "Yes, in another department in the company" and "Yes, in my own department". Individuals with downsizing in their own department were asked three additional questions ('In conjunction with the latest downsizing, consider these claims': 1 My wishes and input have been taken into account in the planning and execution of the change', 2 'I have had the opportunity to speak with my closest leader about the consequences that the change will have for me', and 3 'I have received the necessary training in relation to new tasks and roles'). Response alternatives were based on a five-level Likert scale ('Totally disagree'=1; 'Partly disagree'=2; 'In part yes and in part no'=3; 'Partly agree'=4; 'Totally agree'=5). Principal component analysis indicated a single-factor structure, explaining 61% of the variance, with satisfactory internal consistency (Cronbach's alpha=0.65). The responses were averaged to create an index (1−5) and the index further dichotomized to distinguish between employees who rated involvement, information, and support during downsizing as sufficient (>= 3) and insufficient (< 3). For respondents who replied "not applicable" or had missing values on one or two questions, we imputed values from the questions that were answered. Respondents who replied "not applicable" or had missing values on all three questions were excluded from analyses involving this variable.

## Mental distress symptoms

The Hopkins Symptoms Checklist (HSCL) is an extensive questionnaire of symptoms reported by patients treated for depression and anxiety [31]. A shorter 25 question version (HSCL-25) show high concordance (86,7%) with clinician's rating of patient's level of mental distress [32], which again correlates with even shorter versions (HSCL-5 and 10) [33]. The five questions comprising HSCL-5 were included in each survey, measuring whether respondents experienced specific symptoms in the last two weeks, including feeling fearful, nervousness or shakiness inside, feeling hopeless about the future, feeling blue, and worrying too much, with a 4-level scoring from 'not afflicted' to 'very much afflicted'. By scoring each question from 1 = 'not afflicted' to 4 = 'very much afflicted', the mean score (range 1 - 4) was obtained. A dichotomous variable using a suggested cut-off of ≥ 2.0 categorised respondents with or without mental distress [33].

## Covariates and other variables

The questionnaire provided data on job tenure, job insecurity (risk of job loss within three years), intentions to leave now or in the next years, leader with personnel responsibility, and workplace sector (private or public). The National Education database provided data on attained education, categorised into primary (1 – 2), secondary (3 – 5) and tertiary (6 – 8) level. Respondents verified or corrected registry data on company industry, categorised by the first digit (9 groups) of the NACE classification.

## Statistical analysis

First, we compare the study population to the source population, and compare the downsizing exposure groups, using logistic and linear regression models for categorical and continuous variables, respectively.

Next, we quantify the risk of incident mental distress at the follow-up survey according to downsizing exposure groups using logistic regression. Odds ratios (ORs) along with 95% confidence intervals (CIs) and p-values are reported for: 1 any exposure to downsizing versus no downsizing, 2 downsizing in another department or own department versus no

downsizing, 3 downsizing in another department, own department (sufficient involvement, information, and support), and own department (insufficient involvement, information, and support) versus no downsizing, and 4 a direct comparison of the insufficient versus the sufficient group in a subgroup of employees with downsizing in own department.

Two models were fitted: the first included age and sex, the second also included educational attainment, job tenure, job insecurity, personnel responsibility, turnover intentions, and company industry, sector, and downsizing in the three years before the baseline survey. A random intercept to handle dependency between sets of observations from the same individual was not included, as results were not materially different from a model without it.

To estimate the number of incident cases of mental distress prevented in a counterfactual scenario where downsizing either did not occur, or if exposed employees all rated involvement, information, and support as sufficient, we calculated population attributable fractions (PAF) using the graphPAF package in R statistical software and the function PAF_calc_discrete. PAF's are reported as percentage with 95% CI.

## Results

### Selection of the study population

We describe the selection process starting with all respondents that attended at least one survey (32,752 nested in 22,573 individuals) (Fig 1), wherein we identified 7,954 individuals who attended two consecutive surveys, either the 2013 and

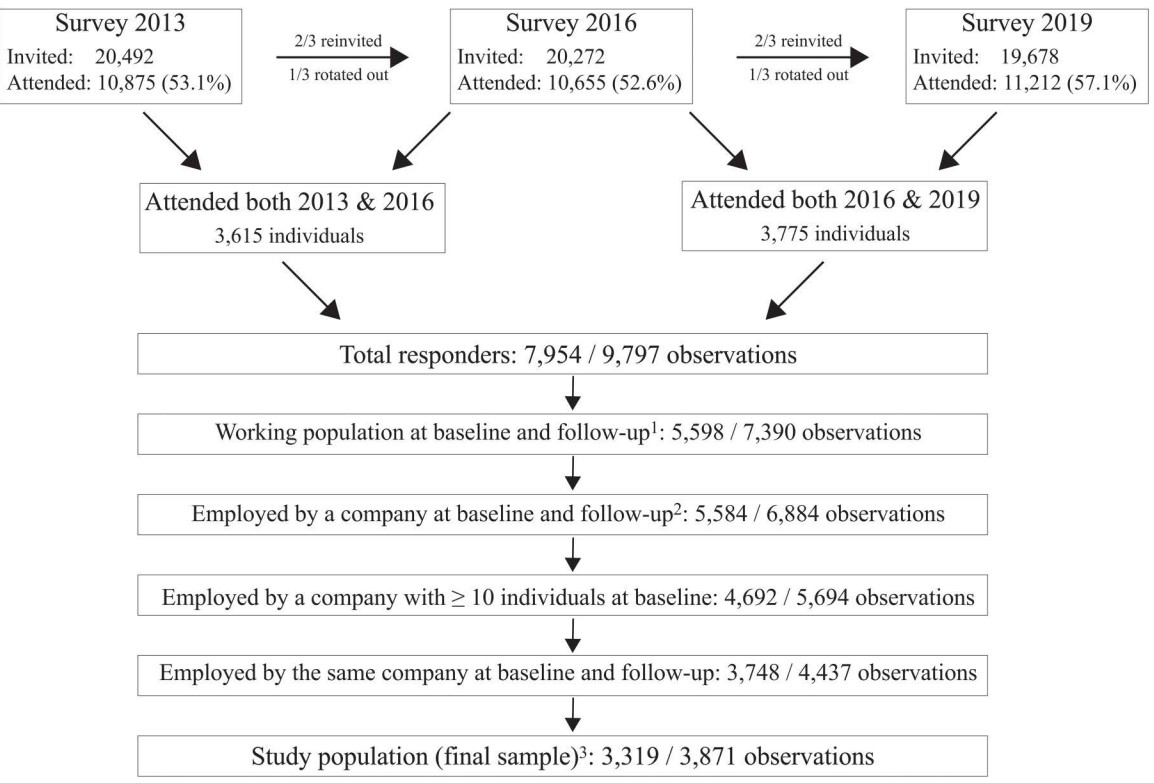

**Fig 1. Flow-chart describing the selection of the study population.** [1] Working was defined as in paid work or temporarily absent from such work at the time of the baseline survey, and employed was defined on a question of whether the person was employed or self-employed. [2] Self-employed individuals were not eligible for the study. [3] Not eligible: Mental distress at the baseline survey (n = 246). Excluded for missing data: downsizing exposure (n = 64); education (n = 24); mental distress at follow-up survey (n = 11); tenure (n = 6); private or public sector (n = 3); industry (n = 46); downsizing in the period prior to the at baseline survey (n = 51); job insecurity (n = 43); turnover intention (n = 72).

PLOS Mental Health

2016 surveys and/or the 2016 or 2019 surveys, corresponding to 9,797 sets of observations from these individuals. Furthermore, 4,437 observations nested within 3,748 individuals were continuously employed by the same company at the baseline and follow-up survey, and the company size was at least 10 people at baseline. After excluding respondents with mental distress at the baseline survey ($n=246$) and those with missing values, the final study population comprised 3,871 observations nested within 3,319 individuals.

## Descriptive statistics of the source and study population

Company employed workers who attended at least one of the three surveys comprised the source population (24,027 respondents nested in 16,847 individuals). Compared to the source population, employees in the study population ($n=3,871$) were on average 5.5 years older, had obtained longer education, were less likely to work in the private sector, had a longer job tenure and exhibited a lower prevalence of mental distress. They reported lower levels of job insecurity and were less likely to be looking for a new job in the future at the baseline survey. The distribution of women and men, as well as the percentage with a leader position, was similar (Table 1).

Among the 3,871 employees, 2,571 (66,4%) were not exposed to downsizing during the three years leading up to the follow-up survey. In total 1,300 (33,6%) were exposed to downsizing occurring either in another ($n=516$) or their own ($n=784$) department. Among employees exposed to downsizing in their own department, 74% ($n=580$) rated involvement, information, and support during downsizing as sufficient and 19% ($n=149$) as insufficient, while the remaining 7% answered 'not relevant' ($n=53$), or had missing values ($n=2$).

Employees exposed to downsizing were overall more frequently men, employed in private companies, and had longer average job tenure in the company compared to unexposed employees (Table 1). They were also more likely to have experienced downsizing in their own department in the 3 years preceding the baseline survey, and at that time, job insecurity and turnover intentions were higher compared with unexposed employees. The prevalence of mental distress was higher among employees exposed to downsizing in their own department and who rated involvement, information, and support as insufficient, compared with the other groups.

## Association of downsizing exposure with the risk of mental distress

Table 2 presents the risk of incident mental distress in the downsizing exposure groups. We report estimates from the comprehensive model. Employees exposed to downsizing had a higher risk of mental distress in comparison with unexposed employees, with an OR (95% CI, p-value) of 1.60 (1.16, 2.22, $p=0.004$). Using these estimates, we calculated that 15.8% (2,3%, 28.4%), corresponding to 12 of 78 cases in the exposed group, could be attributed to downsizing.

Analyses differentiating employees according to proximity to downsizing within the company, showed a higher risk among employees exposed to downsizing in their own department, and an attenuated and less clear difference in risk among employees exposed to downsizing in another department, compared with unexposed employees. Corresponding ORs were 1.79 (1.24, 2.56, $p=0.002$) and 1.31 (0.80, 2.06, $p=0.262$), respectively. The PAR (95% CI) was 3,2% (-2,1%, 11.1%) for downsizing in another department and 12,6% (0,9%, 24.6%) in own department. The latter estimate corresponds to 7 of the 53 cases of mental distress.

Employees exposed to downsizing in their own department also reported involvement, information, and support during downsizing. In comparison with unexposed employees, the risk of mental distress was higher among employees who rated it as insufficient, but not different among those who rated it as sufficient. Corresponding ORs were 5.13 (95% CI 3.04, 8.39, p<0.001) and 1.28 (0.81, 1.96, $p=0.276$), respectively. In a subgroup analysis, the OR was 4.17 (2.26, 7.68, p<0.001) for the direct comparison of the insufficient versus the sufficient group. Using this latter estimate, we estimated that 34.4% (16.5%, 49.9%), corresponding to 8 of the 24 incident cases of mental distress in the group, could be attributed to their ratings of involvement, information, and support as insufficient.

**Table 1. Descriptive characteristics of participants in the source population (n=24,027) and in the study population (n=3,871 observations).**

| | Source population[1] | Study population (conditional on attendance in two consecutive surveys) | | | | | | |
|---|---|---|---|---|---|---|---|---|
| | All | All | P-value[2] | Downsizing in current company in the past 3 years | | | | P-value[2] |
| | | | | No | Yes, another department | Yes, own department | | |
| | | | | | | Sufficient involvement, information, and support[3] | Insufficient involvement, information, and support[3] | |
| N | 24,027 | 3871 | | 2,571 | 516 | 580 | 149 | |
| *Follow-up survey measurements* | | | | | | | | |
| Age, years, mean (SD) | 42.6 (12.9) | 48.1 (10.6) | <0.001 | 48.0 (10.8) | 48.1 (10.2) | 48.1 (10.3) | 49.3 (10.6) | 0.269 |
| Women, n (%) | 11,517 (47.9) | 1,818 (47.0) | 0.298 | 1,302 (50.6) | 190 (36.8) | 248 (42.8) | 58 (38.9) | <0.001 |
| Education, mean (SD), range 1–3 | 2.33 (0.72) | 2.49 (0.62) | <0.001 | 2.49 (0.63) | 2.51 (0.60) | 2.49 (0.61) | 2.43 (0.62) | 0.459 |
| Private company (%) | 15,009 (62.8) | 2,114 (54.6) | <0.001 | 1,242 (48.3) | 362 (70.2) | 366 (63.1) | 104 (69.8) | <0.001 |
| Job tenure, years, mean (SD) | 9.9 (9.4) | 14.8 (9.5) | <0.001 | 14.5 (9.4) | 15.6 (9.9) | 15.2 (9.6) | 15.4 (10.1) | 0.019 |
| Mental distress[4] | 1,725 (7.2) | 186 (4.8) | <0.001 | 108 (4.2) | 25 (4.8) | 29 (5.0) | 24 (16.1) | <0.001 |
| *Baseline survey measurements* | | | | | | | | |
| Downsizing in own department past 3 years | 3,521 (15.2) | 625 (16.1) | 0.149 | 279 (10.9) | 117 (22.7) | 178 (30.7) | 38 (25.5) | <0.001 |
| Leader with personnel responsibility | 4,525 (18.8) | 773 (20.0) | 0.077 | 489 (19.0) | 114 (22.1) | 136 (23.4) | 25 (16.8) | 0.101 |
| Job insecurity | 2,441 (10.3) | 282 (7.3) | <0.001 | 120 (4.7) | 59 (11.4) | 69 (11.9) | 23 (15.4) | <0.001 |
| Looking for new job or turnover intentions | 4,360 (28.4) | 674 (17.4) | <0.001 | 409 (15.9) | 92 (17.8) | 133 (22.9) | 31 (20.8) | <0.001 |

[1]The source population is comprised of respondents in the 2013, 2016 and 2019 surveys (nested in 16,847 individuals) that were working at the time of the survey (or temporarily absent) and employed by a company. Descriptive statistics are average values in the three surveys, in contrast to the values for the study population, where the values are from a baseline or follow-up survey.

[2]P-values are from unadjusted comparisons of mean values or proportions, as appropriate.

[3]Among 784 respondents reporting downsizing in their own department, only 729 could be further categorised according to the questions on information, involvement and support, while 53 replied "not applicable" and 2 had missing values.

[4]Mental distress is defined as a mean score per question ≥ 2.00 on the five-question version of the Hopkins Symptom Checklist.

**Table 2. Risk of developing incident mental distress from the baseline to the follow-up survey according to downsizing exposure, proximity to downsizing in the company, and rating of involvement, information, and support during downsizing, based on 3,871 respondents.**

| *Downsizing in current company in the past 3 years, measured at the follow-up survey* | Risk of developing mental distress (HSCL-5 mean score per question ≥ 2.00) from baseline to the follow-up survey | |
| --- | --- | --- |
| | Model 1[1] | Model 2[1] |
| Study sample | OR (95%CI) | OR (95%CI) |
| No (n = 2,571) | *1.00* | *1.00* |
| Yes (n = 1,300) | **1.57 (1.16 − 2.13), p = 0.003*** | **1.60 (1.16 − 2.22), p = 0.004*** |
| Yes, in another department (n = 516) | 1.26 (0.79 − 1.95), p = 0.307 | 1.31 (0.80 − 2.06), p = 0.262 |
| Yes, own department (n = 784) | **1.78 (1.26 − 2.49), p = 0.001*** | **1.79 (1.24 − 2.56), p = 0.002*** |
| Sufficient involvement, information, and support (n = 580)[2] | 1.27 (0.82 − 1.91), p = 0.268 | 1.28 (0.81 − 1.96), p = 0.276 |
| Insufficient involvement, information, and support (n = 149)[2] | **5.02 (3.03 − 8.04), p < 0.001*** | **5.13 (3.04 − 8.39), p < 0.001*** |
| Subgroup analysis | | |
| Sufficient involvement, information, and support (n = 580)[2] | *1.00* | *1.00* |
| Insufficient involvement, information, and support (n = 149)[2] | **3.98 (2.21 − 7,15), p < 0.001*** | **4.17 (2.26 − 7.68), p < 0.001*** |

*p* < 0.05 = *

Based on four comparisons: 1 any exposure to downsizing versus no downsizing, 2 downsizing in another department and own department versus no downsizing, 3 downsizing in another department, own department (sufficient involvement, information, and support), and own department (insufficient involvement, information, and support) versus no downsizing, and 4 insufficient versus sufficient involvement, information, and support during downsizing limited to the subgroup reporting downsizing within their own department.

Model 1 was adjusted for age and sex. Model 2 included age and sex, along with educational attainment of the respondent, job tenure in the company, industry and sector (private or public) of the company. Additionally, the following variables measured at the baseline were included: downsizing in current company during the past three years, job insecurity, turnover intentions, and whether the respondent reported holding a leadership position with personnel responsibility.

[2]Excluding 53 respondents that answered, 'not applicable' and 2 with missing values on questions rating the downsizing process.

## Discussion

We studied the association between self-assessed company downsizing and the risk of incident mental distress among downsizing survivors in a study population derived from three iterations (2013, 2016, 2019) of a representative survey of the general working population in Norway. One-third of employees reported working in organizations that had downsized in the past three years, a prevalence comparable to findings from European survey data [24]. This indicates that exposure to downsizing is a common and often unavoidable part of the working life, with the potential to affect a large part of the workforce, as organizations frequently use it not only for survival but also for growth [24].

Employees exposed to downsizing had a higher risk of incident mental distress compared to those unexposed. One interpretation is that the association reflects a causal relationship: exposure to downsizing, even without job loss, can reduce employee well-being and increase symptoms of depression and anxiety through other mechanisms. These mechanisms may include increased job insecurity [25] and adverse changes to the work environment due to restructuring, such as higher physical and psychological demands [2]. In our study, 6.2% (n = 78) of the 1245 employees exposed to downsizing developed mental distress, and 12 of these cases were attributable to downsizing. Extrapolated to Norway's 2.7 million current employees [34], this corresponds to over 8,000 new cases of distress. Although this number should be interpreted with caution and in the light of the limitations of this study, it illustrates the potential mental health burden associated with downsizing.

To understand the implications of increased levels of mental distress in the group exposed to downsizing, we apply a staging model that recognises mental health as a continuum — from well-being to subclinical symptoms to mental disorders [35]. While mental well-being may be defined as an asymptomatic state without distress, a mental disorder is characterised

by the presence of severe and specific symptoms that cause distress, loss of function and reduced work ability. In the workplace, this may result in a need for sick leave, support and accommodation. According to theory, individuals with subclinical symptoms, such as those classified as having mental distress in our study, may also experience functional impairments with implications for working life. The elevated risk of mental distress among individuals exposed to downsizing therefore suggest that downsizing can affect work ability, although we did not measure this in the current study.

The finding reinforces prior research on the negative effects of downsizing on employee well-being. Studies from Finland and the U.S. have linked downsizing to increased psychotropic drug use and mental health-related prescriptions [14,17]. However, findings from Norway have been mixed. One registry-based study examining personnel reductions and self-reported distress found no association [20], whereas another study reported increased mental distress among those who reported job terminations in their workplace [19]. Even though downsizing may serve organizational goals, its negative impact on employee mental health warrants attention, particularly regarding measures to mitigate its effects.

Not all employees exposed to downsizing share the same risk of developing mental distress. We found a clearer and more pronounced association among employees experiencing downsizing in their own department. In contrast, the association was attenuated among employees exposed to downsizing in another department. This aligns with findings from a U.S. study, where employees personally threatened by layoffs or being laid-off and rehired showed greater job insecurity, depressive symptoms, and intent to quit than those indirectly exposed from colleagues or friends being laid off [36]. Our findings support the inference that proximity to downsizing intensifies its impact and this finding may help organizations prioritize preventive measures for those most affected.

A key question is whether the negative effects of downsizing can be mitigated. Our results show that employees who experienced downsizing in their own department and rated involvement, information, and support as insufficient had a higher risk of mental distress, with approximately 34,4% of the mental distress cases in this group attributable to their rating. In contrast, those who found these aspects sufficient did not have a higher risk compared to unexposed employees. This suggests that organisations can reduce the negative impact of downsizing by attending to these factors during the planning and execution of downsizing. A healthier change process, defined in part by the same factors, was associated with higher control and support and lower levels of stress among employees in another study [37], and in cross-sectional data from European surveys, both information and involvement were identified as mitigating factors on the relationship of downsizing and restructuring with well-being [24]. A systematic review of the literature supports that employees' perceptions of the restructuring process–particularly factors such as transparent communication and procedural justice – are linked to better outcomes [8]. Lastly, research suggest that strategic downsizing, as opposed to reactive downsizing, leads to better outcomes, perhaps by giving organisations more time to plan and implement socially responsible downsizing and restructuring [38].

Policies and laws, such as those outlined for the European Union [24], emphasize employer responsibilities during downsizing and restructuring. These regulations establish minimum requirements for a fair process, including timely information and employee involvement, primarily aimed at protecting those who lose their job. While adhering to these requirements may help organizations avoid legal issues related to unfair processes, they might not be sufficient to prevent negative effects on the well-being of remaining employees or to ensure that the benefits of downsizing are not undermined by increased mental distress [8]. A more holistic approach to downsizing may be necessary – one that also includes sufficient information and employee involvement throughout the process for all employees at risk of negative effects, and adequate training when restructuring leads to new job roles or tasks. This approach could also be relevant in smaller organizations or in cases where only a few employees lose their job, even though existing policies on information and involvement do not apply to the same extent.

### Strength and limitations

The study population was derived from representative surveys of the Norwegian working population, which strengthen the external validity of the study. Younger employees were underrepresented, but we argue that findings can be generalised

to employees in the general working population. The distribution of measured covariates differed between the downsizing exposure groups, and although adjusting for them had minimal impact, residual confounding cannot be ruled out as a source of bias.

Downsizing was measured by asking employees whether their current company had undertaken downsizing in the past three years. It implies some degree of action, but personnel reduction is not measured explicitly. We argue that this measurement might be more sensitive to personnel reduction of a certain size or severity, such as layoffs, as downsizing is a term often used in the context of layoffs, and less sensitive to other modes of personnel reduction that objectively are less severe, including natural attrition.

The size of downsizing on company level may influence how strongly or likely it is that downsizing affects the average employee, and the extent to which the company provides sufficient employee involvement, information and support. It was not possible to calculate net personnel reduction from questionnaire data or obtain it from registry data, but individual level exposure severity was indicated by the question on proximity to downsizing within the company, and we argue that this is a more sensitive measurement.

## Conclusion

Employees exposed to downsizing had a higher risk of mental distress compared to unexposed employees, highlighting a concerning consequence of a widely used organizational strategy in both the private and public sectors. However, our findings suggest that this risk is not uniform. Employees more directly affected by downsizing experienced a stronger impact, and importantly, those who rated involvement, information, and support as sufficient did not show an increased risk. This underscores the potential for organizations to mitigate the negative effects of downsizing through proactive and transparent change management strategies. By prioritizing communication, employee involvement, and adequate support, companies may not only reduce the mental health risks associated with downsizing but also foster a more resilient and sustainable work environment.

## Acknowledgments

We thank the participants in the Survey of Living-Working Conditions. Their willingness to contribute to research enabled this study. We thank Ståle Østhus for input on methodology.

## Author contributions

**Conceptualization:** Eirik Degerud, Andrea Rørvik Marti, Tom Sterud.

**Formal analysis:** Eirik Degerud, Tom Sterud.

**Writing – original draft:** Eirik Degerud.

**Writing – review & editing:** Andrea Rørvik Marti, Tom Sterud.

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
