## [Decision Letter · Decision Letter 0]

PMEN-D-24-00417

Association of company downsizing with the risk of mental distress among employees in the Norwegian working population.

PLOS Mental Health

Dear Dr. Degerud,

Thank you for submitting your manuscript to PLOS Mental Health. After careful consideration, we feel that it has merit but does not fully meet PLOS Mental Health’s publication criteria as it currently stands. Therefore, we invite you to submit a revised version of the manuscript that addresses the points raised during the review process.

We look forward to receiving your revised manuscript.

Kind regards,

Martin Mabunda Baluku, Ph.D.

Academic Editor

PLOS Mental Health

Journal Requirements:

Additional Editor Comments (if provided):

Reviewers' comments:

Reviewer's Responses to Questions

**Comments to the Author**

1. Does this manuscript meet PLOS Mental Health’s publication criteria ? Is the manuscript technically sound, and do the data support the conclusions? The manuscript must describe methodologically and ethically rigorous research with conclusions that are appropriately drawn based on the data presented.

Reviewer #1: Yes

Reviewer #2: Partly

2. Has the statistical analysis been performed appropriately and rigorously?

Reviewer #1: Yes

Reviewer #2: Yes

3. Have the authors made all data underlying the findings in their manuscript fully available (please refer to the Data Availability Statement at the start of the manuscript PDF file)?

Reviewer #1: Yes

Reviewer #2: Yes

4. Is the manuscript presented in an intelligible fashion and written in standard English?

Reviewer #1: Yes

Reviewer #2: Yes

5. Review Comments to the Author

Reviewer #1: Rational of the study on the topic could have been further elaborated, and the importance of the study could have been mentioned clearly. Selection of participants in the study, if presented graphically it would have been easier to understand and visualize the sample selection process of the participants. In descriptive statistics section of result, findings could have been presented in fraction or percentage rather than presenting in numbers because it is easier to visualize results. Odds ratio could have been presented to one decimal point. In result table 2, statistically significant p-values could have been highlighted with "*" and bolded as it becomes eye catchy for readers. In discussion section, the probable factors that have favored the association of different factors of downsizing with poor mental health outcomes could have been added for each statistically significant association. The factors associated with, poor mental health outcomes as a result of downsizing could have been compared with similar studies of downsizing and its association with mental health in different context. After the completion of study, recommendation must have been made to protect mental health of employees even though employees go through the downsizing in the organizations. How the finding of the study could be helpful to promote and protect mental health of the employees where downsizing is frequent? How can this study will be helpful for the future researchers?

Reviewer #2: Thank you for giving me the opportunity to review this manuscript.

I thank the authors for a well written and clear manuscript. The methods are clear, the analysis fits the study objectives and the results are easy to follow. While the paper is beautifully written, it can be improved with some minor revisions below.

1. Downsizing and mental distress is widely covered in literature, the authors need to provide a clear and unique contribution of their study to literature. What is new about this study that is not known? This clarity will make the study more valid.

2. Line 45-47 the message is not clear, revision might be needed

3. While the authors have done a descent job in most parts of the manuscript, the discussion has not been given sufficient attention. As it presently stands, the discussion sounds more of a repetition of results not a deep delve into the meaning of results and an elaborate arrangement of arguments. This section needs to be thoroughly improved. Furthermore, the discussion still lacks depth and a holistic representation of findings.

6. PLOS authors have the option to publish the peer review history of their article (what does this mean? ). If published, this will include your full peer review and any attached files.

**Do you want your identity to be public for this peer review?** For information about this choice, including consent withdrawal, please see our Privacy Policy .

Reviewer #1: No

Reviewer #2: No

---

## [Decision Letter · Decision Letter 1]

PMEN-D-24-00417R1

Association of company downsizing with the risk of mental distress among employees in the Norwegian working population.

PLOS Mental Health

Dear Dr. Eirik Degerud,

Thank you for submitting your manuscript to PLOS Mental Health. After careful consideration, we feel that it has merit but does not fully meet PLOS Mental Health’s publication criteria as it currently stands. Therefore, we invite you to submit a revised version of the manuscript that addresses the points raised during the review process.

Please submit your revised manuscript by May 15, 2025. If you will need more time than this to complete your revisions, please reply to this message or contact the journal office at mentalhealth@plos.org. Please include the following items when submitting your revised manuscript:

We look forward to receiving your revised manuscript.

Kind regards,

Martin Mabunda Baluku, Ph.D.

Academic Editor

PLOS Mental Health

Journal Requirements:

Additional Editor Comments (if provided):

Reviewers' comments:

Reviewer's Responses to Questions

**Comments to the Author**

1. If the authors have adequately addressed your comments raised in a previous round of review and you feel that this manuscript is now acceptable for publication, you may indicate that here to bypass the “Comments to the Author” section, enter your conflict of interest statement in the “Confidential to Editor” section, and submit your "Accept" recommendation.

Reviewer #1: All comments have been addressed

Reviewer #2: (No Response)

2. Does this manuscript meet PLOS Mental Health’s publication criteria ? Is the manuscript technically sound, and do the data support the conclusions? The manuscript must describe methodologically and ethically rigorous research with conclusions that are appropriately drawn based on the data presented.

Reviewer #1: Yes

Reviewer #2: Partly

3. Has the statistical analysis been performed appropriately and rigorously?

Reviewer #1: Yes

Reviewer #2: Yes

4. Have the authors made all data underlying the findings in their manuscript fully available (please refer to the Data Availability Statement at the start of the manuscript PDF file)?

Reviewer #1: Yes

Reviewer #2: Yes

5. Is the manuscript presented in an intelligible fashion and written in standard English?

Reviewer #1: Yes

Reviewer #2: Yes

6. Review Comments to the Author

Reviewer #1: The revised manuscript is well written and all the comments has been addressed properly. The manuscript can be further processed towards the publication.

Reviewer #2: I thank the authors for paying careful attention to the comments and addressing them in an intelligible fashion.

I however invite them to make some small revisions to help strength the manuscript.

1. Line 273, sub-heading "Comparison with other studies and interpretation" should be removed because it is redundant. This is what the discussion section is all about, thus, the heading "Discussion" is sufficient.

2. The discussion has greatly improved but could still be made better. After reporting what was found, the authors have a responsibility to interpret what such a finding means, even in the Norway context if possible. We need to know what such results mean before contrasting with other studies. This has not come out clearly. Take an example. Line 281, "Employees exposed to downsizing had a higher risk of incident mental distress" before relating to literature, how do authors interpret this result? what does it mean? once the meaning is clearly elaborated then contrasting with other studies can come in. This alignment helps the reader understand how authors made meaning out of their findings.

7. PLOS authors have the option to publish the peer review history of their article (what does this mean? ). If published, this will include your full peer review and any attached files.

**Do you want your identity to be public for this peer review?** For information about this choice, including consent withdrawal, please see our Privacy Policy .

Reviewer #1: No

Reviewer #2: No

---

## [Decision Letter · Decision Letter 2]

Association of company downsizing with the risk of mental distress among employees in the Norwegian working population.

PMEN-D-24-00417R2

Dear Dr. Eirik Degerud,

We are pleased to inform you that your manuscript 'Association of company downsizing with the risk of mental distress among employees in the Norwegian working population.' has been provisionally accepted for publication in PLOS Mental Health.

Best regards,

Martin Mabunda Baluku, Ph.D.

Academic Editor

PLOS Mental Health

Reviewer Comments (if any, and for reference):

Reviewer's Responses to Questions

**Comments to the Author**

1. If the authors have adequately addressed your comments raised in a previous round of review and you feel that this manuscript is now acceptable for publication, you may indicate that here to bypass the “Comments to the Author” section, enter your conflict of interest statement in the “Confidential to Editor” section, and submit your "Accept" recommendation.

Reviewer #1: All comments have been addressed

Reviewer #2: All comments have been addressed

2. Does this manuscript meet PLOS Mental Health’s publication criteria ? Is the manuscript technically sound, and do the data support the conclusions? The manuscript must describe methodologically and ethically rigorous research with conclusions that are appropriately drawn based on the data presented.

Reviewer #1: Yes

Reviewer #2: Yes

3. Has the statistical analysis been performed appropriately and rigorously?

Reviewer #1: I don't know

Reviewer #2: Yes

4. Have the authors made all data underlying the findings in their manuscript fully available (please refer to the Data Availability Statement at the start of the manuscript PDF file)?

Reviewer #1: Yes

Reviewer #2: Yes

5. Is the manuscript presented in an intelligible fashion and written in standard English?

Reviewer #1: Yes

Reviewer #2: Yes

6. Review Comments to the Author

Reviewer #1: All the comments has been well addressed and can be moved further toward publication.

Reviewer #2: The manuscript has greatly improved after the revisions

7. PLOS authors have the option to publish the peer review history of their article (what does this mean? ). If published, this will include your full peer review and any attached files.

**Do you want your identity to be public for this peer review?** For information about this choice, including consent withdrawal, please see our Privacy Policy .

Reviewer #1: No

Reviewer #2: No
